# The Role of Osmolytes and Membrane Lipids in the Adaptation of Acidophilic Fungi

**DOI:** 10.3390/microorganisms11071733

**Published:** 2023-07-01

**Authors:** Elena A. Ianutsevich, Olga A. Danilova, Olga A. Grum-Grzhimaylo, Vera M. Tereshina

**Affiliations:** 1Winogradsky Institute of Microbiology, Research Center of Biotechnology of the Russian Academy of Sciences, 33, bld. 2 Leninsky Ave., 119071 Moscow, Russia; noitcelfer@mail.ru (O.A.D.); v.m.tereshina@inbox.ru (V.M.T.); 2White Sea Biological Station, Faculty of Biology, Lomonosov Moscow State University, 1–12 Leninskie Gory, 119234 Moscow, Russia; olgrgr@wsbs-msu.ru; 3Laboratory of Genetics, Plant Sciences Group, Wageningen University, Droevendaalsesteeg 1, 6708 PB Wageningen, The Netherlands

**Keywords:** extremophilic fungi, *Phlebiopsis gigantea*, *Mollisia*, trehalose, phosphatidic acids

## Abstract

Acidophiles maintain near-neutral intracellular pH using proton pumps. We have suggested the protective role of osmolytes and membrane lipids in the adaptation to an acidic environment. Previously we have observed, for the first time, high levels of trehalose in acidophilic basidiomycete *Sistotrema brinkmannii*. Here, we have studied the composition of both osmolytes and membrane lipids of two more acidophilic fungi. Trehalose and polyols were among the main osmolytes during growth under optimal conditions (pH 4.0) in basidiomycete *Phlebiopsis gigantea* and ascomycete *Mollisia* sp. Phosphatidic acids, phosphatidylethanolamines, phosphatidylcholines, and sterols, were predominant membrane lipids in both fungi. *P. gigantea* had a narrow optimum of growth at pH 4.0, resulting in a sharp decline of growth rate at pH 2.6 and 5.0, accompanied by a decrease in the number of osmolytes and significant changes in the composition of membrane lipids. In contrast, *Mollisia* sp. had a broad optimal growth range (pH 3.0–5.0), and the number of osmolytes either stayed the same (at pH 6.0) or increased (at pH 2.6), while membrane lipids composition remained unchanged. Thus, the data obtained indicate the participation of osmolytes and membrane lipids in the adaptation of acidophilic fungi.

## 1. Introduction

Fungi are thought to play a key role in the functioning of extreme ecosystems [1]. The study of extremophiles makes it possible to identify biomolecules’ adaptation mechanisms and properties that allow them to master extreme habitat niches. This is important for understanding the biochemical foundations of life on Earth and astrobiology and ecology—preserving biocenoses when faced with abrupt climate changes and increased anthropogenic influence [2]. Extremophiles are essential as a source of unusual bioactive secondary metabolites for biotechnological and medicinal purposes (with antimicrobial, cytotoxic, anti-inflammatory, antioxidant and anti-allergic activities) [3].

Extremophilic fungi are divided into several groups according to temperature (thermophiles, psychrophiles), osmotic factor (halophiles, xerophiles), pH range (alkaliphiles, acidophiles), and pressure (barophiles) [4]. Naturally, acidic habitats (pH 3.0–4.0) are common, including soils, lakes, swamps, and peatlands [5]. Extremely acidic habitats (pH < 3.0) are also quite widespread and can be natural (terrestrial hydrothermal vents, volcanic lakes) or of anthropogenic origin (acidic dumps of coal mines, mine and industrial wastewater, drainage).

Most fungi prefer near-neutral environmental conditions (pH 6.0–7.0). Acidophiles are fungi capable of growing up to pH 1.0, but unable to grow under neutral conditions, whereas acidotolerants can grow under acidic, neutral or even slightly alkaline conditions [1]. Acidophilic yeasts isolated from habitats with pH 1–3 maintain an optimal cellular pH of 4.5–5.5 [6]. For the acidophilic filamentous fungus, *Acidomyces acidophilus* pH 3.0–5.0 is the growth optimum, but it retains its ability to grow even at pH 1.0 [1]. In addition to bacteria and archaea, eukaryotes, such as algae, protozoa and fungi, are also present in acidic habitats with low pH [7]. In natural hyperacidic habitats (pH < 3.0), bacteria, fungi, algae and protozoa form biofilms, which become the site of metals and minerals deposition and serve as a substrate for subsequent populations of microorganisms [8,9].

Acidic environmental conditions change the membrane potential, and affect the absorption of substrates, the function of proteins, and the toxicity of metal ions [10]. One of the main mechanisms of adaptation to acidic environments is maintaining a neutral intracellular pH by using proton pumps that provide the efflux of hydrogen ions out of the cell [11]. In fungi, the intracellular pH regulation system includes vacuolar-type ATPases (V-ATPase) and a P-type proton pump Pma1, acting with many other transporters [12]. Current research of the acidophilic fungus *Acidiella bohemica* genome showed that under the stress of low pH 1.5, encoded KUP system potassium uptake protein, cation transporters, H^+^ transporting ATPases, Na^+^/H^+^ exchanger, and several symporters [13]. Transcriptomic data indicated that H^+^ transporting ATPases may be the most active. The *A. bohemica* genome contained several DNA and protein repair genes: dnaK, dnaJ, ASF1, hscB, GRPE, PSMG3, and HSP90A. Among them, dnaK and HSP90A have the highest transcriptional activity under pH stress. It has been identified that transcription factor Nrg1 in the basidiomycetous fungus *Ustilago maydis* plays a significant role in several cellular processes, including the response to acid pH, morphogenesis and virulence [14].

Numerous studies of the fungal response to various stress factors have shown the importance of changes in the osmolytic and membrane systems in adaptation [15,16]. Osmolytes include low-molecular-weight organic compounds of various chemical natures that protect cells against the stress effects. Osmolytes are compatible solutes with neutralizing and cytoprotective properties, protecting macromolecules and cell membranes [17,18]. In fungi, osmolytes are represented by a number of polyols and trehalose, sometimes also by the amino acid proline [19]. All osmolytes are multifunctional compounds. Thus, disaccharide trehalose has protective, reserve, antioxidant, regulatory, and chaperone functions [20,21,22,23]. Polyols have osmoprotective and antioxidant properties, regulating the redox balance of the cell [19,24,25].

It has been established that psychrophilic, halophilic, and xerophilic fungi use osmolytes and changes in the membrane lipids composition to adapt to extreme conditions [26,27,28]. We have demonstrated, for the first time, the importance of osmolyte trehalose and phosphatidic acids (PA) in the composition of membrane lipids for fungal thermophilia [29,30,31] and alkaliphilia [32,33].

We have previously suggested that osmolytes and membrane lipids may be involved in the adaptation to acidic conditions by protecting the cytoplasmic membrane from an aggressive external environment [34]. Our assumption was confirmed by the example of the acidophilic basidiomycete fungus *Sistotrema brinkmannii*, which accumulated a large amount of trehalose in the mycelium, while sphingolipids (SL), phosphatidylethanolamines (PE), phosphatidylcholines (PC), and PA were predominant among membrane lipids. To further prove the role of the osmolytic and membrane systems in the adaptation of acidophilic fungi, it was necessary to confirm the data obtained using other fungi of various systematic positions.

In this work, we aimed to obtain the physiological characteristic of the acidophilic fungi *Phlebiopsis gigantea* and *Mollisia* sp., and to study the composition of osmolytes and membrane lipids in growth dynamics under optimal conditions and at different pH.

## 2. Materials and Methods

### 2.1. Objects of Study and Cultivation Protocol

Basidiomycete fungus *Phlebiopsis gigantea* (Fr.) Jülich (*Phanerochaetaceae*, *Polyporales*, *Incertae sedis*, *Agaricomycetes*, *Agaricomycotina*, *Basidiomycota*, *Fungi*); was isolated from peat from a depth of 0.5 m of the transitional aapa-type peatland, identified by ITS rDNA region and deposited at GenBank with the accession number JQ780612 [31].

Ascomycete fungus *Mollisia* sp. (*Mollisia* (Fr.) P. Karst., (*Mollisiaceae*, *Helotiales*, *Leotiomycetidae*, *Leotiomycetes*, *Pezizomycotina*, *Ascomycota*, *Fungi*); was isolated from the deep bottom sediments of the aapa-type oligotrophic peatland (sampling depth 2.3 m); identified by ITS (internal transcribed spacer 1, 5.8S ribosomal RNA gene, and internal transcribed spacer 2) and LSU (large subunit ribosomal RNA gene) regions of rDNA and deposited in GenBank with the accession numbers JX507644 and JX507645, respectively [35].

The fungi were cultivated on a standard agarized medium based on malt extract, 17 g/L (“Condalab”), with citrate phosphate buffer at optimal pH 4.0 (pH tolerance range 2.6–6.0) at an optimum temperature of 24–25 °C during 10–14 days and stored at 5–8 °C. The optimum growth of fungi was determined by the rate of linear growth in surface culture. Agarized media consisted of two components: (1) the citrate-phosphate buffer component with various pH values (2.6, 3.0, 4.0, 5.0, 6.0, 7.0, 7.6) and the nutrient component, containing 34 g/L of the malt extract (Condalab, Madrid, Spain) with 40 g/L of agar. Components (1) and (2) were autoclaved separately at 121 °C for 20 min, cooled down to 55 °C and then mixed in 1:1 ratio under sterile conditions, resulting in the final concentrations of the complete medium: 17 g/L malt extract, 20 g/L agar. The inoculum was grown on an agarized medium on Petri dishes for four days (pH 4.0, 24–25 °C). For plate inoculation, 1 × 1 mm of mycelium was used, cut from the actively growing edge of the colony. The diameter of the colonies was measured in two perpendicular directions every 3–4 days until the colony reached the edges of the plate in one of the Petri dishes. The growth rate was determined for the period of linear growth. By linear growth rate, we mean the rate during the linear phase of growth, when the colony radius increased for the same increments during the equal time intervals, resulting in linear function r(t). We plotted the colony radius on the vertical axis and time on the horizontal axis and used the linear part of the resulting plot to calculate the growth rate. The optimum temperature was determined by measuring the growth rate of the fungus at an optimal pH of 4.0 at temperatures of 17, 20, 22, 25, 27, and 30 °C.

For biochemical studies of lipids and osmolytes in growth dynamics and at different pH values, fungi were pre-grown as follows. The mycelium agar plugs were cut from the parental agar slant culture and used to inoculate several 90 mm Petri dishes with the pH 4.0 medium, further cultivated at 24–25 °C for 7–10 days. The mycelium fragments were then cut from the actively growing edge of the colony and used to inoculate the appropriate number of 90 mm Petri dishes with a cellophane-coated medium. *P. gigantea* was cultivated for 5, 10 and 14 days, and *Mollisia* sp.—for 12, 23, and 35 days under optimal conditions (pH 4.0, 24–25 °C). To investigate the effects of different pH values, fungi were cultivated for 14 days (*P. gigantea*) at pH 2.6, 4.0 and 5.0 and 35 days (*Mollisia* sp.) at pH 2.6, 4.0 and 6.0. The mycelium was carefully separated from the cellophane with a scalpel, weighed, and frozen at −21° C.

### 2.2. Lipids, Carbohydrates and Polyols Analysis

Lipids, carbohydrates and polyols were analysed as described earlier [30]. Briefly, lipids were extracted by the Nichols method [36] with phospholipase-deactivating isopropanol, separated by two-dimensional (polar lipids) [37] or one-dimensional (neutral lipids) [38] thin-layer chromatography (TLC) and quantified using standard compounds with the densitometry method (DENS software, version 5.1.0.2). To study the composition of fatty acids, the polar lipid fraction was isolated using one-dimensional TLC with a solvent system for neutral lipids. The polar lipid spots were scraped out from the origin spots and eluted with a mixture of chloroform: methanol (1:1). The extract was evaporated, and methanolysis was carried out using 2.5% H_2_SO_4_ in methanol for 2 h at 70 °C [39]. The resulting methyl esters were analyzed by gas-liquid chromatography (GLC) on a Kristall 5000.1 gas-liquid chromatograph (Chromatec, Yoshkar-Ola, Russia) with an Optima-240, 60 m × 0.25 µm × 0.25 mm capillary column (Macherey-Nagel GmbH&Co, Düren, Germany). The temperature program was from 130 to 240 °C at a rate of 5–6 °C/min. Identification was made using the Supelco 37 Component FAME Mix mixture of individual fatty acid methyl esters (Supelco, Bellefonte, PA, USA). The degree of unsaturation of the phospholipids (DU) was calculated according to the following equation [40]:DU = 1.0 × (% monoene FA)/100 + 2.0 × (% diene FA)/100 + 3.0 × (% triene FA)/100

To determine the soluble carbohydrate composition of the mycelium, sugars were extracted with boiling water for 20 min four times. Proteins were removed from the resulting total extract [41]. The carbohydrate extract was further purified from charged compounds using a combined column with the Dowex-1 (acetate form) and Dowex 50 W (H^+^) ion exchange resins. Carbohydrate composition was determined by GLC using trimethylsilyl sugar derivatives obtained from the lyophilized extract [42]. The internal standard was α-methyl-D-mannoside (Merck). Chromatography was performed on a Kristall 5000.1 gas chromatograph (Chromatec, Yoshkar-Ola, Russia) with a ZB-5 30 m, 0.32 mm, 0.25 μm capillary column (Phenomenex, Torrance, CA, USA). The temperature was increased from 130 to 270 °C at a rate of 5–6 °C/min. Glucose, mannitol, arabitol, inositol, and trehalose (Sigma, St. Louis, MO, USA) were standards.

### 2.3. Statistical Analysis

The experiments were carried out in triplicate, *n* = 3. The post hoc Dunnett test was used for pairwise comparison between the control (pH 4.0) and pH 2.6, 5.0 (for *P. gigantea*) and pH 2.6, 6.0 (for *Mollisia* sp.). Asterisks (*) show a statistically significant difference (*p* ≤ 0.05). On all graphs, mean values ± SEM (standard error of the mean) are plotted.

## 3. Results

### 3.1. Physiological Characteristics of Fungi

At pH 4.0 on the malt agar media *P. gigantea* forms colonies that are light-yellowish in color, slightly raised, and circular with the whole margin. Mycelia is a uniform, thin, smooth film (Figure 1a). The overall appearance of the colony is similar for pH 2.6 and 5.0.

At pH 4.0 on the malt agar media, *Mollisia* sp. forms colonies that are brown in the center, getting light-beige in color, slightly raised, circular with filamentous margin, mycelia present as uniform thick film, fluffy in the center, smooth towards the edge (Figure 1b). The colony appears drier and powdery at pH 2.6 and moister and wrinkly at pH 6.0.

A study of growth rates at different pH showed that both fungi belong to obligate acidophiles since their growth optimum is at pH 4.0, while there is no growth at pH 7.0 (Figure 2a). However, *P. gigantea* has a higher growth rate (3.2 mm/day) with a pronounced peak at pH 4.0, while at pH 3.0 and 5.0, there is a threefold decrease in the growth rate. In contrast, the growth rate of *Mollisia* sp. is twice as low at an optimal pH of 4.0 (1.6 mm/day), but the range of its active growth is much wider (3.0–5.0). In relation to temperature, both fungi are mesophiles with a broad optimum at 20–27 °C, (Figure 2b).

### 3.2. Composition of Carbohydrates and Polyols (CaP) of Cytosol in the Mycelium of Fungi under Optimal Conditions in Growth Dynamics and at Different pH

The CaP of *P. gigantea* is mainly represented by polyol arabitol, carbohydrates glucose, and trehalose, with trace amounts of glycerol, erythritol, inositol, and mannitol (Figure 3). In growth dynamics, the amount of CaP doubles and reaches 7.5% of the dry weight of mycelium due to an increase in the amount of trehalose and arabitol. Trehalose (44–54% of the total CaP) and arabitol (32–39%) are predominant at all stages of growth, while the level of glucose decreases from 24 to 7%. Compared to the optimal pH value of 4.0, there is a threefold decrease in the amount of CaP at pH 2.6 due to a decrease in the levels of arabitol by one and a half times and, especially, trehalose by eight times. At the pH value of 5.0, a twofold drop in CaP is observed, where the amount of trehalose is reduced in half, and arabitol—one and a half times. At pH 2.6, arabitol is the predominant component of CaP (68% of the total), and at pH 5.0—trehalose (54% of the total).

In *Mollisia* sp. CaP is represented mainly by mannitol and trehalose, while glycerol, erythritol, glucose, and arabitol are in trace amounts (Figure 4). Over time the amount of CaP decreases by 1.75 times due to a 2.5-fold decrease in the level of trehalose, while the level of mannitol does not change significantly. In growth dynamics, the proportion of mannitol increases from 55 to 70% of the total CaP, and the proportion of trehalose decreases from 45 to 30%. Compared with the optimal pH value of 4.0, the amount of CaP increases two-fold at pH 2.6 due to the four times increase of the trehalose and one-and-a-half times increase of the mannitol levels, resulting in equalizing their amounts. In contrast, at pH 6.0, no significant changes are observed changes in either the amount or composition of CaP compared to the optimal conditions.

### 3.3. The Composition of Membrane Lipids under Optimal Conditions in the Growth Dynamics and at Different pH Values

The total amount of *P. gigantea* membrane lipids reaches 6% of dry weight and does not change significantly during growth and with the pH changes (Table 1). Membrane lipids are represented by phospholipids (85% of the total), sterols (10%) and sphingolipids (5%). The predominant components at all stages of growth are PA, PE, PC and sterols (Figure 5). Over time the proportion of PE increases against the background of a decrease in PA. Cultivation at pH 2.6 leads to an increase in the relative amount of PC, while the proportion of PA decreases 2-fold. In contrast, at pH 5.0, the percentage of PA is doubled due to the decrease in PE and PC.

In *Mollisia* sp., membrane lipids are represented by phospholipids (70% of the total), sterols (25% of the total) and sphingolipids (5% of the total). The amount of lipids does not exceed 3% of the dry weight and does not change depending on cultivation time and pH (Table 1). At pH 4.0, the main components are PA, sterols, PE, and PC (Figure 6). In growth dynamics, a decrease in the proportion of PA and a slight increase in the proportions of sterols and PC are observed. A change in the pH of the medium to 2.6 or 6.0 does not result in any noticeable changes in the ratio of the membrane lipids components.

### 3.4. The Composition of Storage Lipids under Optimal Conditions in the Growth Dynamics and at Different pH Values

Storage lipids of the *P. gigantea* make up about 6% of the dry weight at all stages of growth and different pH (Table 1) and are mainly represented by monoacylglycerols (MAG), diacylglycerols (DAG), triacylglycerols (TAG) and free fatty acids (FFA) in all experimental variants (Figure 7). In growth dynamics, only a slight increase in the proportions of MAG and DAG against the background of a decrease in FFA can be noted. However, at pH 2.6 and 5.0, the TAG proportion increases two-fold due to a decrease in the MAG.

In *Mollisia* sp., the amount of storage lipids reaches 16% of dry weight and does not change depending on the growth phase and pH of the medium (Table 1). The composition of storage lipids is heavily dominated by TAG, accounting for 70–90% of the total (Figure 8). Over time, at pH 4.0, the percentage of TAG decreases to 65% of the total, and the proportion of DAG increases up to 15%. Alterations of the pH to 2.6 and 6.0 increase the relative amount of TAG to 90% of the total, while the proportion of DAG declines slightly at pH 2.6.

### 3.5. The Fatty Acids Composition of Polar Lipids under Optimal Conditions in the Growth Dynamics and at Different pH Values

The fatty acids composition of *P. gigantea* polar lipids is dominated by linoleic acid (C18:2)—70% of the total, followed by palmitic acid (C16:0)—20% of the total, with trace amounts of the remaining fatty acids (Table 2). Neither culture age nor changes in pH alter the fatty acid composition, leading to the constant degree of unsaturation of approximately 1.48.

*Mollisia* sp. features three main fatty acids in the composition of polar lipids: linoleic (18:2), oleic (C18:1) and palmitic (16:0) (Table 2). In growth dynamics, at pH 4.0, a slight decrease in the degree of unsaturation can be observed due to an increase in the proportion of oleic acid against the background of a decrease in linoleic acid. At pH 6.0, they are no noticeable changes in the degree of unsaturation, while acidic conditions of pH 2.6 cause its slight decline due to a decrease in the proportion of linoleic acid against the background of an increase in the proportion of myristoleic acid (C14:1).

## 4. Discussion

According to their physiological characteristics, both fungi under study belong to obligate acidophiles (optimal pH 4.0, no growth at pH 7.0) and mesophiles (optimal temperature 20–27 °C) (Figure 2). We have previously suggested that osmolytes may be involved in the adaptation to acidic conditions by protecting the cytoplasmic membrane from the acidic environment [34] since it is known that osmolyte trehalose can protect not only the macromolecules but also the membranes of the cell [20,21,22,23]. For a long time, it remained unclear how intracellular trehalose can protect the cytoplasmic membrane from the outside. Recent studies using Saccharomyces cerevisiae have shown that there is a trehalose transporter Agt1 [43]. Under heat shock, Agt1-containing vesicles fused with the membrane to transport part of the cytosolic trehalose to the outside. The lack of Agt1 reduced cell thermotolerance and increased lipid peroxidation. To study the mechanisms of fungal adaptation to an acidic environment, we simultaneously studied the composition of osmolytes and membrane lipids and their interrelations. Our results obtained on the example of two fungi of different systematic positions, including a previous study of acidophile *S. brinkmannii* [34], convincingly demonstrate that the osmolytic system is involved in adaptation to acidic environmental conditions. Thus, in the basidiomycete fungus, *P. gigantea* trehalose is the main osmolyte at all stages of growth, and its share increases from 44 to 54% (Figure 3). A distinctive feature of *P. gigantea* is a narrow-pronounced peak of maximum growth rate (3.2 mm/day) at pH 4.0. Cultivation of the fungus at pH 2.6 and 5.0, with the growth rate being sharply reduced by 5 and 3 times, respectively, results in a drop in the level of trehalose, which indicates the specificity of trehalose synthase of this fungus and explains its inability to adapt to the changes of the pH. Previously we have obtained a similar result for the acidophilic fungus *S. brinkmannii* with a narrow peak of optimal growth at pH 3.0–4.0 [34]. In contrast, *Mollisia* sp. has a broad range of optimal pH 3.0–5.0, but the growth rate at pH 4.0 is half that of *P. gigantea*. Notably, the growth rate decreases threefold at pH 6.0 and only by 30% at pH 2.6. Trehalose, along with mannitol, is predominant at all stages of growth, but its amount decreases over time (Figure 4). However, cultivation under acidic conditions (pH 2.6) results in an increase of the trehalose level (4-fold) and mannitol (1.5-fold), which is consistent with the ability to grow under these conditions, implying the adaptive role of osmolytes. Overall, the data obtained indicate the characteristic specificity of trehalose synthase of the studied fungi. In *P. gigantea*, this enzyme is active only at pH 4.0, while *Mollisia* sp. has a wider pH range from 2.6 to 4.0.

Acidophilic archaea membrane has specific tetraether lipids with ether bonds resistant to acid hydrolysis, in contrast to ester bonds of eukaryotic membrane lipids [44]. These lipids help to reduce the permeability of the membrane to protons. Here we observed no unusual membrane lipids in acidophilic fungi. We also did not find an increase in the proportion of phospholipids in the composition of membrane lipids, which was noted in the example of the yeast *Rhodotorula glutinis* under acidic conditions compared to near-neutral ones [45]. Previously, a significant increase in the proportion of non-bilayer PA in the composition of membrane lipids under the action of heat shock in three mesophilic fungi has been demonstrated for the first time [46], as well as a high percentage of these phospholipids in thermophilic fungi [30], in alkaliphiles [32,33] and xerophiles [47]. Taken together with our previous work [34], PA is one of the main components of the membrane in the three studied acidophilic fungi, which indicates their special role in extremophilia, but its function remains unclear.

PA is a multifunctional metabolite with structural, signaling, and regulatory roles [48,49]. It is the central intermediate in the synthesis of membrane and storage lipids. Dephosphorylation of PA by PA phosphatase produces DAG, from which the PC and PE phospholipids are formed via the Kennedy pathway. In addition, the acylation of DAG leads to the formation of storage TAG lipids. PA is a precursor for membrane phospholipids synthesized via the liponucleotide intermediate CDP-DAG (cytidine-diphosphate diacylglycerol) [50]. PA/DAG ratio plays a regulatory role in lipid signaling, lipid droplet formation, vesicular trafficking, phospholipid synthesis, gene expression. The small head group of the PA, negative charge and phosphomonoester group determine its specific interactions with proteins as regulatory and stabilizing compounds [51]. It is known that, depending on conditions, PA can exhibit both bilayer and non-bilayer lipid properties [52]. Thus, at neutral pH and in the absence of divalent ions, PA exhibits the properties of bilayer lipids, but under weakly acidic conditions and in the presence of ions, for example, in the Golgi apparatus, they form type II micelles [53]. It is believed that the ability of phosphatidic acids to aggregate leads to the formation of microdomains, forming membrane curvatures, which is the first stage of the formation of vesicles that participate in transport from the Golgi apparatus and EPR, endo- and exocytosis. Thus, PA has emerged as a class of new lipid mediators influencing membrane structure, dynamics, and protein interactions [54]. Taking into account the fact that PA is the main or one of the main phospholipids in the acidophiles and other extremophilic fungi previously studied by us [30,32,47,55], PA structural role may be assumed, indicating a special membrane structure of extremophiles.

In the first acidophilic fungus, we studied—*S. brinkmannii*—an unusual profile of membrane lipids with predominant SL (up to 60% of the total), followed by PE, PA and sterols, was observed [34]. In the growth dynamics, the proportions of PE and PA increased 2-fold, while the proportion of SL was halved. It remained unclear whether the high percentage of SL is characteristic of fungal acidophilia or is a specific feature of the particular fungus. Here, another basidiomycete fungus—*P. gigantea*—was studied, revealing SL as a minor component, with predominant PA, PE, PC, and sterols (Figure 5). In growth dynamics, the proportion of PA decreases against the background of an increase in PE. At pH 2.6, the growth rate of the fungus is reduced 3-fold, accompanied by a decrease in the proportion of PA and an increase in PC, and at pH 5.0, the growth rate decreases less and an increase in the proportion of PA against the background of a decrease in PE is observed. Notably, the proportion of sterols remains 10% during growth and at different pH. In ascomycete *Mollisia* sp., as well as in *P. gigantea*, PA, PE, PC, and sterols are predominant in the composition of membrane lipids (Figure 6). Taken together, these data indicate the importance of PA for fungal extremophilia. Unlike *P. gigantea*, in the ascomycete fungus *Mollisia* sp. with a wide pH growth range, we observe very small changes in the composition of membrane lipids at different pHs. At the same time, it should be noted that this fungus has a large number of storage TAG (about 16% of the dry weight), which are considered a source of fatty acids for phospholipids [56]. In both fungi, under the influence of unfavorable pH, an increase in the amount and percentage of TAG among storage lipids was noted. A study of the fatty acids composition of polar lipids for both fungi revealed insignificant changes depending on the stage of growth and pH, suggesting that the mechanism of changing the degree of unsaturation is not involved in the adaptation to the pH factor.

Previously, we suggested an interrelation between the osmolyte and membrane systems of fungal cells, and it was shown by the example of the haloalkalitolerant fungus *Emericellopsis alkalina* that the osmolyte system plays a key role in adapting to alkaline conditions and a high concentration of NaCl, while changes in the composition of membrane lipids are insignificant [55]. It has also been established that heat shock causes more significant rearrangements of the membrane lipids than in wild strains [56]. Here, we have also observed a similar pattern. Thus, *Mollisia* sp. shows no decrease in the level of osmolytes at pH 6.0 and a significant increase in the level of trehalose and mannitol at pH 2.6, accompanied by small changes in the composition of membrane lipids and their fatty acids. In contrast, in *P. gigantea*, we observed a sharp drop in the level of osmolytes, especially trehalose, under unfavorable pH conditions and simultaneously significant fluctuations in the composition of membrane lipids.

## 5. Conclusions

The study of acidophilic fungi *P. gigantea* and *Mollisia* sp., including the previously obtained data for *S. brinkmannii* [34], confirmed our assumption about the significant role of osmolytes in the adaptation of acidophilic fungi to acidic environmental conditions. Trehalose and polyols were among the main osmolytes in the mycelium of all studied fungi during growth under optimal conditions (pH 4.0). On the example of these three fungi, it can also be stated that one of the main phospholipids of acidophiles is PA, suggesting their essential function in fungal extremophilia, along with our previous data for thermophilic, alkaliphilic, and xerophilic fungi. Basidiomycete *P. gigantea*, with a narrow optimum at pH 4.0, had a sharp decrease in growth rate at pH 2.6 and 5.0, accompanied by a decrease in the number of osmolytes and significant changes in the composition of membrane lipids. In contrast, *Mollisia* sp., with a broad optimal pH range (3.0–5.0), demonstrated no changes in the membrane lipids composition, while the amount of osmolytes either increased at pH 2.6 or remained constant at pH 6.0. The mechanism of altering the unsaturation degree of membrane phospholipids is not involved in the adaptation of acidophiles to the pH factor. The data obtained has also confirmed our assumption about the interrelation of changes in the osmolytic and membrane systems under the influence of stress factors.

## Figures and Tables

**Figure 1 microorganisms-11-01733-f001:**
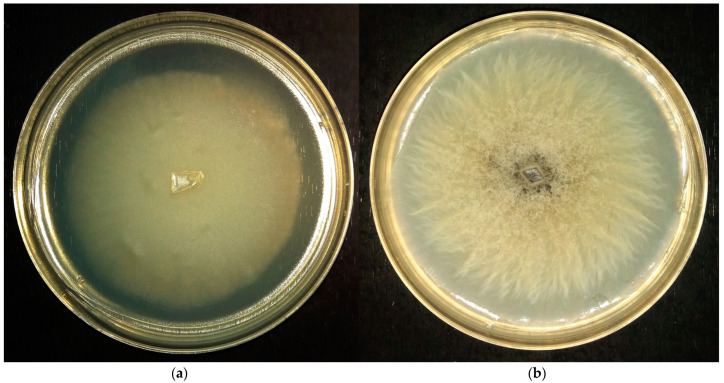
Typical colonies of acidophilic micromycetes *P. gigantea* (**a**) and *Mollisia* sp. (**b**) grown at optimal conditions (pH 4.0, 24–25 °C) for 12 days (**a**) or 33 days (**b**).

**Figure 2 microorganisms-11-01733-f002:**
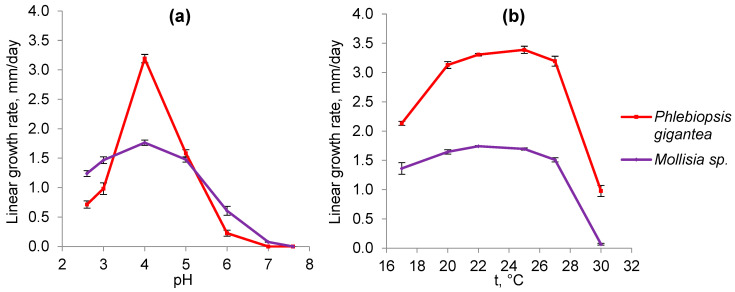
The effect of pH (**a**) and temperature (**b**) on the growth rate of *P. gigantea* and *Mollisia* sp.

**Figure 3 microorganisms-11-01733-f003:**
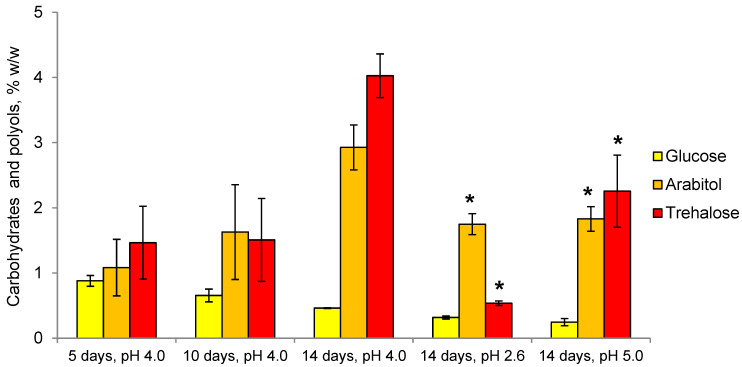
Carbohydrates and polyols of *P. gigantea* in growth dynamics and at different pH values. Means ± SEM are plotted, *n* = 3, * *p* ≤ 0.05. SEM—standard error of the mean.

**Figure 4 microorganisms-11-01733-f004:**
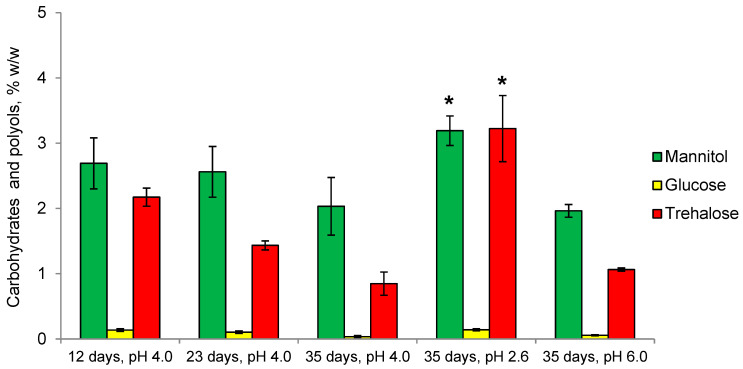
Carbohydrates and polyols of *Mollisia* sp. in growth dynamics and at different pH values. Means ± SEM are plotted, *n* = 3, * *p* ≤ 0.05.

**Figure 5 microorganisms-11-01733-f005:**
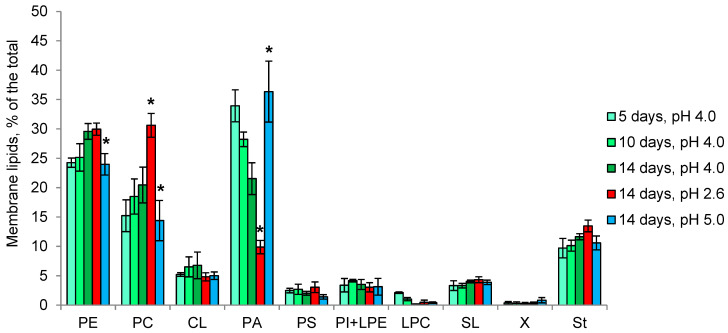
The profile of *P. gigantea* membrane lipids in growth dynamics and at different pH values. PE—phosphatidylethanolamines, PC—phosphatidylcholines, CL—cardiolipins, PA—phosphatidic acids, PS—phosphatidylserines, PI—phosphatidylinositols, LPE—lysophosphatidylethanolamines, LPC—lysophosphatidylcholines, SL—sphingolipids, X—unidentified lipid, St—sterols. Means ± SEM are plotted, *n* = 3, * *p* ≤ 0.05.

**Figure 6 microorganisms-11-01733-f006:**
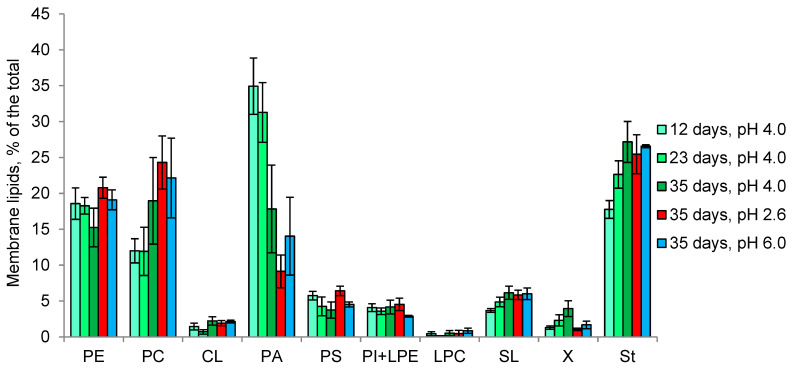
The profile of *Mollisia* sp. membrane lipids in growth dynamics and at different pH values. Means ± SEM are plotted, *n* = 3.

**Figure 7 microorganisms-11-01733-f007:**
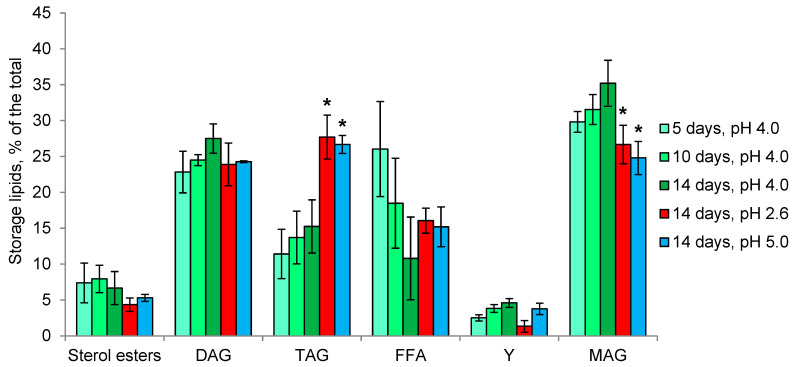
Storage lipids profile of *P. gigantea* in growth dynamics and at different pH values. DAG—diacylglycerols, TAG—triacylglycerols, FFA—free fatty acids, Y—unidentified lipid, MAG—monoacylglycerols. Means ± SEM are plotted, *n* = 3, * *p* ≤ 0.05.

**Figure 8 microorganisms-11-01733-f008:**
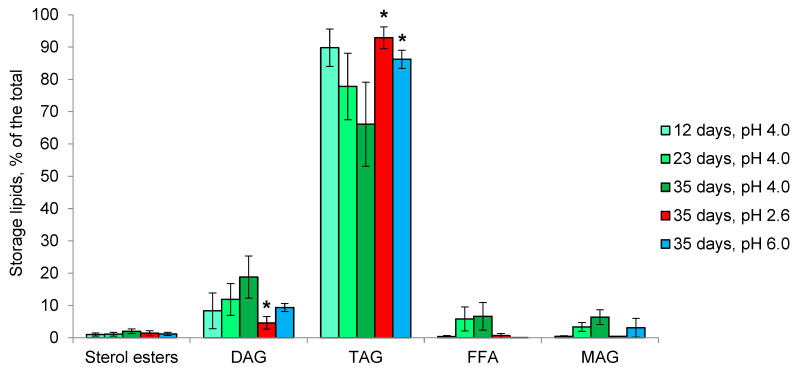
Storage lipids profile of *Mollisia* sp. in growth dynamics and at different pH values. Means ± SEM are plotted, *n* = 3, * *p* ≤ 0.05.

**Table 1 microorganisms-11-01733-t001:** Amounts of membrane and storage lipids (% of dry weight) of *P. gigantea* and *Mollisia* sp.

*P. gigantea*	Five Days, pH 4.0	10 Days, pH 4.0	14 Days, pH 4.0	14 Days, pH 2.6	14 Days, pH 5.0
**Membrane lipids**	7.1 ± 0.99	5.44 ± 0.2	4.39 ± 0.35	3.46 ± 0.35	5.24 ± 0.29
**Storage lipids**	6.7 ± 0.84	6.12 ± 1.18	5.53 ± 0.67	5.1 ± 0.28	5.02 ± 0.95
***Mollisia* sp.**	**12 days, pH 4.0**	**23 days, pH 4.0**	**35 days, pH 4.0**	**35 days, pH 2.6**	**35 days, pH 6.0**
**Membrane lipids**	3.01 ± 0.18	2.44 ± 0.15	2.13 ± 0.25	2.5 ± 0.19	2.17 ± 0.26
**Storage lipids**	13.3 ± 1.34	16.61 ± 2.71	16.45 ± 3.64	16.4 ± 0.42	15.81 ± 0.85

**Table 2 microorganisms-11-01733-t002:** Fatty acids of polar lipids in growth dynamics and at different pH values (% of the total). Means ± SEM are displayed, *n* = 3, **p* ≤ 0.05. SEM—standard error of the mean.

Fatty Acids	*P. gigantea*					*Mollisia* sp.				
	5 Days, pH4	10 Days, pH4	14 Days, pH4	14 Days, pH2.6	14 Days, pH5.0	12 Days, pH 4.0	23 Days, pH 4.0	35 Days, pH 4.0	35 Days, pH 2.6	35 Days, pH 6.0
**C 14:0 (myristic)**	0.73 ± 0.05	0.63 ± 0.32	0.62 ± 0.31	0.84 ± 0.07	0.72 ± 0.05	0.24 ± 0.24	0.35 ± 0.35	0.31 ± 0.31	0	0
**C 14:1 (myristoleic)**	0.5 ± 0.27	0.33 ± 0.33	0.95 ± 0.49	1.09 ± 0.95	0.67 ± 0.55	1.22 ± 1.22	2.1 ± 2.1	1.02 ± 1.02	13.1 ± 4.08 (*)	10.69 ± 5.35 (*)
**C 15:0 (pentadecylic)**	1.16 ± 0.18	2.37 ± 0.52	2.27 ± 0.44	0.5 ± 0.39	1.14 ± 0.51	0.1 ± 0.1	0.13 ± 0.13	0	0.34 ± 0.17	0
**C 16:0 (palmitic)**	19.47 ± 1.12	19.04 ± 1.48	19.37 ± 0.51	18.42 ± 0.85	18.87 ± 0.32	19.84 ± 1.86	21.68 ± 1.25	20.64 ± 1.23	20.18 ± 2.19	14.98 ± 0.53 (*)
**C 16:1 (palmitoleic)**	0.97 ± 0.16	0.53 ± 0.28	0.84 ± 0.28	1.01 ± 0.16	1 ± 0.15	0.14 ± 0.14	0.17 ± 0.17	0.09 ± 0.09	0	0
**C 17:0 (margaric)**	0.72 ± 0.19	0.65 ± 0.34	0.92 ± 0.13	0.49 ± 0.1	0.5 ± 0.19	1.65 ± 0.86	2.25 ± 1.14	2.17 ± 1.23	1.23 ± 0.62	1.34 ± 0.67
**C 17:1 (heptadecenoic)**	0.4 ± 0.03	0.3 ± 0.15	0.25 ± 0.12	0.32 ± 0.02	0.33 ± 0	0.23 ± 0.23	1.27 ± 0.87	1.31 ± 1.03	0	0
**C 18:0 (stearic)**	1.18 ± 0.02	1.2 ± 0.4	1.18 ± 0.41	0.94 ± 0.06	1.14 ± 0.1	1.65 ± 0.21	1.14 ± 0.12	1.3 ± 0.28	1.48 ± 0.3	1.38 ± 0.13
**C 18:1n9c (oleic)**	3.75 ± 0.26	3.12 ± 0.1	4.08 ± 1.57	4.54 ± 0.41	5 ± 0.37	19.88 ± 2.75	26.1 ± 1.45	32.22 ± 2.55	32.83 ± 2.64	28.09 ± 1.42
**C 18:2n6c (linoleic)**	71.05 ± 1.58	71.84 ± 2.6	69.46 ± 1.19	71.81 ± 2.03	70.58 ± 0.81	54.4 ± 3.94	44.7 ± 0.98	40.94 ± 2.18	30.85 ± 0.33 (*)	43.51 ± 3.94
**C 18:3n3 (linolenic)**	0.06 ± 0.06	0	0.07 ± 0.07	0.05 ± 0.05	0.06 ± 0.06	0.65 ± 0.05	0.12 ± 0.12	0	0	0
**Degree of unsaturation**	1.48 ± 0.03	1.48 ± 0.05	1.45 ± 0.01	1.51 ± 0.03	1.48 ± 0.01	1.32 ± 0.05	1.19 ± 0.04	1.17 ± 0.05	1.08 ± 0.01	1.26 ± 0.04

## Data Availability

Data are contained within the article.

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
