# Peer review of "The Role of Osmolytes and Membrane Lipids in the Adaptation of Acidophilic Fungi"

_microorganisms, 2023, doi:10.3390/microorganisms11071733_

Round 1
Reviewer 1 Report
Overview and general recommendation
The article is concerned with the osmolytes role in the adaptation of acidophilic fungi to acidic environments. The findings suggest that one of the main phospholipids of acidophilic fungi are phosphatidic acids (PA) which have an important function in fungal extremophilia. This paper fits within the scope of Microorganisms. Nevertheless, the article is recommended for acceptance after minor revision.
Some comments to authors:
Line 90: Please add a description of the abbreviation ITS.
Line 94: Please add a description of the abbreviation LSU.
Line 125: Please add a description of the abbreviation of GLC.
Large subunit ribosomal (LSU) DNA is widely used in fungal phylogenetics, anyway I wanted to know why you have chosen this method.
Which method you have used for accurate taxonomic classifications (BLAST, Metagenomic analyzer, MEGAN, SAP, etc.)?
Will it be possible to provide Specific PCR primers for the amplification of Nuclear ITS and LSU rDNA?
Will it be possible to provide a schematic diagram of large subunit ribosomal DNA (LSU rDNA)? (as supplementary material)
It would be advantageous to provide a Figure of the morphograph of fungi used.
Table 2. It’s difficult to follow, please add the name of fatty acids in brackets close to the abbreviations.
It would be better to add more recent references.
Author Response
Overview and general recommendation
The article is concerned with the osmolytes role in the adaptation of acidophilic fungi to acidic environments. The findings suggest that one of the main phospholipids of acidophilic fungi are phosphatidic acids (PA) which have an important function in fungal extremophilia. This paper fits within the scope of Microorganisms. Nevertheless, the article is recommended for acceptance after minor revision.
Some comments to authors:
Comment 1
Line 90: Please add a description of the abbreviation ITS.
Response
The abbreviation ITS (internal transcribed spacer 1, 5.8S ribosomal RNA gene, and internal transcribed spacer 2) was added to the text.
Comment 2
Line 94: Please add a description of the abbreviation LSU.
Response
The abbreviation LSU (large subunit ribosomal RNA gene) was added to the text.
Comment 3
Line 125: Please add a description of the abbreviation of GLC.
Response
The description of GLC was added: gas-liquid chromatography
Comment 4
Large subunit ribosomal (LSU) DNA is widely used in fungal phylogenetics, anyway I wanted to know why you have chosen this method.
Response
We used LSU DNA for Mollisia sp., because there was not enough information in the GenBank to identify this fungus by ITS.
Comment 5
Which method you have used for accurate taxonomic classifications (BLAST, Metagenomic analyzer, MEGAN, SAP, etc.)?
Response
We have used BLAST.
Comment 6
Will it be possible to provide Specific PCR primers for the amplification of Nuclear ITS and LSU rDNA?
Response
Yes, it’s possible, but this information is given in the article we link to in the article (Grum-Grzhimaylo et al., 2016)
Comment 7
Will it be possible to provide a schematic diagram of large subunit ribosomal DNA (LSU rDNA)? (as supplementary material)
Response
This is public information on the Internet. In our opinion, it is not necessary here, since the work is devoted to cellular mechanisms not related to LSU DNA.
Comment 8
It would be advantageous to provide a Figure of the morphograph of fungi used.
Response
We did not take photographs of fungal colonies, since the article is devoted to biochemical changes, but the description of the colonies is given in the text.
Comment 9
Table 2. It’s difficult to follow, please add the name of fatty acids in brackets close to the abbreviations.
Response
The names of fatty acids were added to the Table 2.
Comment 10
It would be better to add more recent references.
Response
We added a few more references, please, kindly see the manuscript with tracked changes.
Reviewer 2 Report
Manuscript submitted by Ianutsevich et al. address an important question related to the adaptation of extremophiles. However, I found the manuscript structure with a very low quality. The manuscript needs a lot of corrections in all the sections. Some of the major/minor comments are provided below:
The appropriate sampling design is missing.
Line no. 90 & 95: Author’s provided the accession no. of deposited DNA sequence. What about the accession of fungal strains. If they are deposited in any culture collection? Provide details.
At Line no. 97. Author’s mentioned optimum pH = 4 but not the range of pH tolerance. Please provide the pH range also.
At Line no. 99: Author’s have mentioned the agarized media at pH<3-4; but at such low pH how agar helped? I didn’t understand. Please elaborate.
Methodology related to the GLC is not provided. Please provide sample preparation and experimentation in details.
At Line no. 133: What is SEM? Please provide full form
Figure 1. is of very poor quality. Please improve and follow the identical fonts. How the growth was measured in pH <3-4 where the media is not solidified. Because at pH <3-4 agar does not solidify. Provide details. In addition to graph, pictures of the growing cultures at different temperature and pH are also required. The pictures are the real result for the Figure 1. Please provide the details about the error bars. If it denotes SE or SD? How do you define linear growth? Please explain.
Section 3.2: How the carbohydrate, sugars etc are measured. Please improve the quality of figure 2.
Improve the quality of all figures and tables.
Results are not discussed well, please improve the discussion with the relevant recent references.
Check all the references and follow an identical pattern. For example, at Line no. 250, reference is not in the correct format.
Author Response
Manuscript submitted by Ianutsevich et al. address an important question related to the adaptation of extremophiles. However, I found the manuscript structure with a very low quality. The manuscript needs a lot of corrections in all the sections. Some of the major/minor comments are provided below:
Comment 1
The appropriate sampling design is missing.
Response
We have added the following section to the Materials and Methods:
For biochemical studies of lipids and osmolytes in growth dynamics and at different pH values, fungi were pre-grown as follows. The mycelium agar plugs were cut from the parental agar slant culture and used to inoculate several 90 mm Petri dishes with the pH 4.0 medium, that was further cultivated at 24–25 °C for 7–10 days. The mycelium fragments then were cut from the actively growing edge of the colony and used to inoculate the appropriate number of 90 mm Petri dishes with a cellophane-coated medium.
We have also added several clarifications to the Materials and Methods section, according to further comments.
Comment 2
Line no. 90 & 95: Author’s provided the accession no. of deposited DNA sequence. What about the accession of fungal strains. If they are deposited in any culture collection? Provide details.
Response
At present, we have provided strains for deposit to the All-Russian Collection of Microorganisms — VKM. Strain numbers will be obtained after checking their purity within 2–3 months.
Comment 3
At Line no. 97. Author’s mentioned optimum pH = 4 but not the range of pH tolerance. Please provide the pH range also.
Response
We added the following to the Materials and Methods:
The fungi were cultivated on a standard agarized medium based on malt extract, 17 g/L (“Condalab”), with citrate phosphate buffer at optimal pH 4.0 (pH tolerance range 2.6–6.0) at an optimum temperature of 24–25°C during 10-14 days and stored at 5-8°C.
Comment 4
At Line no. 99: Author’s have mentioned the agarized media at pH<3-4; but at such low pH how agar helped? I didn’t understand. Please elaborate.
Response
Indeed, if the medium with pH<4.0 is sterilized, then agar hydrolysis occurs and the medium does not solidify. We sterilize the buffer and the wort agar components separately and then combine them both warm (55°C). In this case, agar hydrolysis does not occur and the medium solidifies. We have added the details of the medium preparation to the «Materials and Methods» section.
Agarized media consisted of two components: (1) the citrate-phosphate buffer component with various pH values (2.6, 3.0, 4.0, 5.0, 6.0, 7.0, 7.6) and the nutrient component, containing 34 g/L of the malt extract (Condalab) with 40 g/L of agar. Components (1) and (2) were autoclaved separately at 121 °C for 20 min, cooled down to 55°C and then mixed in 1:1 ratio under sterile conditions, resulting in the final concentrations of the complete medium: 17 g/L malt extract, 20 g/L agar.
Comment 5
Methodology related to the GLC is not provided. Please provide sample preparation and experimentation in details.
Response
We have added the description of quantitative analysis of carbohydrates and polyols, as well as fatty acids to the «Materials and Methods» section.
For carbohydrates and polyols:
To determine the soluble carbohydrate composition of the mycelium, sugars were extracted with boiling water for 20 min four times. Proteins were removed from the resulting total extract (Somogyi 1945). The carbohydrate extract was further purified from charged compounds using a combined column with the Dowex-1 (acetate form) and Dowex 50 W (H+) ion exchange resins. Carbohydrate composition was determined by GLC using trimethylsilyl sugar derivatives obtained from the lyophilized extract (Brobst 1972). The internal standard was α-methyl-D-mannoside (Merck). Chromatography was carried out on a Kristall 5000.1 gas chromatograph (Chromatec, Russia) with a ZB-5 30 m, 0.32 mm, 0.25 μm capillary column (Phenomenex, United States). The temperature was increased from 130 to 270 °C at a rate of 5–6 °C/min. Glucose, mannitol, arabitol, inositol, and trehalose (Sigma, United States) were used as standards.
For fatty acids:
The resulting methyl esters were analyzed by gas-liquid chromatography (GLC) on a Kristall 5000.1 gas-liquid chromatograph (Chromatec, Russia) with an Optima-240, 60 m × 0.25 µm × 0.25 mm capillary column (Macherey-Nagel GmbH&Co, Germany). The temperature program used was from 130 to 240°С at a rate of 5–6 °C/min. Identification was carried out using the Supelco 37 Component FAME Mix mixture of individual fatty acid methyl esters (Supelco, United States). Unsaturation degree of the phospholipids (UD) was calculated according to the following equation (Weete, 1974):
DU = 1.0 × (% monoene FA)/100 + 2.0 × (% diene FA)/100 + 3.0 × (% triene FA)/100
Comment 6
At Line no. 133: What is SEM? Please provide full form
Response
SEM stands for the standard error of the mean. It is used interchangeably with the abbreviation SE. We have added full form to the abbreviation.
Comment 7
Figure 1. is of very poor quality. Please improve and follow the identical fonts. How the growth was measured in pH <3-4 where the media is not solidified. Because at pH <3-4 agar does not solidify. Provide details. In addition to graph, pictures of the growing cultures at different temperature and pH are also required. The pictures are the real result for the Figure 1. Please provide the details about the error bars. If it denotes SE or SD? How do you define linear growth? Please explain.
Response
We have changed the vertical axis font from bold to normal in all figures. We have checked that all figures use identical font — Arial, 10 pt. We have changed the lines and markers on figure1.
Please see our response to comment 4 for the explanation on the agar media at pH <3-4.
We did not take the pictures of fungal colonies, because the article was devoted to biochemical, but not morphological, changes in the fungus mycelium.
The error bars depict SE. We have added the following clarification to the Statistical Analysis section:
On all graphs mean values ±SEM (standard error of the mean) are plotted.
We have added following clarification to the section Materials and Methods:
By linear growth rate we mean the rate during the linear phase of growth, when the colony radius increased for the same increments during the equal time intervals, resulting in linear function r(t). We plotted the colony radius on the vertical axis and time on the horizontal axis and used the linear part of the resulting plot to calculate the growth rate.
Comment 8
Section 3.2: How the carbohydrate, sugars etc are measured. Please improve the quality of figure 2.
Response
See response to the comment 5.
We have also removed the trace polyols from the graph to improve the perception of the picture.
Comment 9
Improve the quality of all figures and tables.
Response
Please see our response to comment 8 and 9. We have also removed trace polyols from figure 3 and unpresented Y from figure 7
Comment 10
Results are not discussed well, please improve the discussion with the relevant recent references.
Response
We added a few more references throughout the text, please, kindly see the manuscript with tracked changes.
Comment 11
Check all the references and follow an identical pattern. For example, at Line no. 250, reference is not in the correct format.
Response
We have checked all references and changed them to the appropriate numerical format.
Reviewer 3 Report
The manuscript presented the characterization of the membrane lipid of 2 fungi strains isolated from peatland. The manuscript was well-written and organized. Some minor points should be revised before publication.
1. Please refer to the following “Nomenclature of Microorganisms” to revise throughout the manuscript.
"Names of all bacterial taxa (kingdoms, phyla, classes, orders, families, genera, species, and subspecies) are printed in italics and should be italicized in the manuscript; strain designations and numbers are not."
https://jb.asm.org/content/nomenclature
2. Line 90: it should be "accession number"
3. The conclusion part which provided a wrap-up of this study should be included.
English presentation is fine, just some minor typos should be checked.
Author Response
The manuscript presented the characterization of the membrane lipid of 2 fungi strains isolated from peatland. The manuscript was well-written and organized. Some minor points should be revised before publication.
Comment 1
Please refer to the following “Nomenclature of Microorganisms” to revise throughout the manuscript.
"Names of all bacterial taxa (kingdoms, phyla, classes, orders, families, genera, species, and subspecies) are printed in italics and should be italicized in the manuscript; strain designations and numbers are not."
https://jb.asm.org/content/nomenclature
Response
We have checked and corrected the text according to the rules.
Comment 2.
Line 90: it should be "accession number"
Response
Corrected.
Comment 3.
The conclusion part which provided a wrap-up of this study should be included.
Response
We have included the the conclusion section to the manuscript, please kindly see the manuscript with tracked changes.
Comments on the Quality of English Language
English presentation is fine, just some minor typos should be checked.
Response
We have checked the manuscript for the typos.
Round 2
Reviewer 2 Report
Ianutsevich et al. have tried to improve the manuscript by incorporating the comments. Still manuscript has to be improved at some points:
1. The quality of the figure doesnt match the standard of the journal, it has to be improved by following the identical pattern. For example, the legend position.
2. Author's have added the description of related to linear growth, this is a significant improvement but no picture of the agar plates/colony is provided. Strain colony pictures are the basis of the measurement of growth.
3. I didnt find any recent relevant reference which is updated in the new version.
Author Response
Ianutsevich et al. have tried to improve the manuscript by incorporating the comments. Still manuscript has to be improved at some points:
Comment 1:
The quality of the figure doesnt match the standard of the journal, it has to be improved by following the identical pattern. For example, the legend position.
Response:
We have changed the sizes of the figures and position of the legends accordingly and we have also removed gridlines for a clearer view.
Comment 2:
Author's have added the description of related to linear growth, this is a significant improvement but no picture of the agar plates/colony is provided. Strain colony pictures are the basis of the measurement of growth.
Response:
We have added Figure 1, demonstrating typical colonies of P. gigantea and Mollisia sp. grown at optimal conditions (pH 4.0, 24–25°C) for 12 days and 33 days accordingly, so that the plates aren’t overgrown.
Comment 3:
I didnt find any recent relevant reference which is updated in the new version.
Response:
We have added several more references in the Discussion section, such as Magalhães et al. 2018; Pokotylo et al. 2018; Carman and Han 2019; Zhukovsky et al. 2019; Zhou et al. 2023.